# Improvement of the Clinical and Psychological Profile of Patients with Autism after Methylcobalamin Syrup Administration

**DOI:** 10.3390/nu14102035

**Published:** 2022-05-12

**Authors:** Adela Čorejová, Tomáš Fazekaš, Daniela Jánošíková, Juraj Repiský, Veronika Pospíšilová, Maria Miková, Drahomíra Rauová, Daniela Ostatníková, Ján Kyselovič, Anna Hrabovská

**Affiliations:** 1Department of Pharmacology, Faculty of Medicine, Slovak Medical University in Bratislava, 833 03 Bratislava, Slovakia; 2Department of Physical Chemistry of Drugs, Faculty of Pharmacy, Comenius University Bratislava, 832 32 Bratislava, Slovakia; tomas.fazekas@uniba.sk; 3Department of Psychology, Faculty of Philosophy and Arts, Trnava University, 918 43 Trnava, Slovakia; janosikovad@centrum.sk (D.J.); juraj.repisky@gmail.com (J.R.); 4Autism Center Andreas in Bratislava, 811 06 Bratislava, Slovakia; vpospisilova@gmail.com; 5Autism Center FRANCESCO in Prešov, 080 01 Prešov, Slovakia; surdamik@hotmail.com; 6Department of Pharmaceutical Analysis and Nuclear Pharmacy, Faculty of Pharmacy, Comenius University Bratislava, 832 32 Bratislava, Slovakia; d.rauova@gmail.com; 7Institute of Physiology, Faculty of Medicine, Comenius University Bratislava, 813 72 Bratislava, Slovakia; daniela.ostatnikova@fmed.uniba.sk; 8Clinical Research Unit, 5th Department of Internal Medicine, Department of Pharmacology and Toxicology, Faculty of Medicine, Comenius University Bratislava, 813 72 Bratislava, Slovakia; kyselovic@fpharm.uniba.sk; 9Department of Pharmacology and Toxicology, Faculty of Pharmacy, Comenius University Bratislava, 832 32 Bratislava, Slovakia

**Keywords:** autism, methylcobalamin, social skills, behavior, communication, cognition, glutathione, oxidative stress

## Abstract

(1) Background: Autism, also known as autism-spectrum disorder, is a pervasive developmental disorder affecting social skills and psychological status in particular. The complex etiopathogenesis of autism limits efficient therapy, which leads to problems with the normal social integration of the individual and causes severe family distress. Injectable methylcobalamin was shown to improve the clinical status of patients via enhanced cell oxidative status and/or methylation capacity. Here we tested the efficiency of a syrup form of methylcobalamin in treating autism. (2) Methods: Methylcobalamin was administered daily at 500 µg dose to autistic children and young adults (*n* = 25) during a 200-day period. Clinical and psychological status was evaluated by parents and psychologists and plasma levels of reduced and oxidized glutathione, vitamin B12, homocysteine, and cysteine were determined before the treatment, and at day 100 and day 200 of the treatment. (3) Results: Good patient compliance was reported. Methylcobalamin treatment gradually improved the overall clinical and psychological status, with the highest impact in the social domain, followed by the cognitive, behavioral and communication characteristics. Changes in the clinical and psychological status were strongly associated with the changes in the level of reduced glutathione and reduced/oxidized glutathione ratio. (4) Conclusion: A high dose of methylcobalamin administered in syrup form ameliorates the clinical and psychological status of autistic individuals, probably due to the improved oxidative status.

## 1. Introduction

Autism (autistic disorder), also known as autism-spectrum disorder, is a pervasive developmental disorder that causes problems with social skills and psychological status, which may prevent the normal social integration of the individual and cause severe family distress. 

The existing autism prevalence estimates are divergent. Recent reports show a 0.62–0.70% prevalence of autism worldwide and a 1–2% prevalence in a large-scale survey [1], while the incidence of occurrence continues to rise [2]. The epidemiology of autism is not strongly affected by geographic region, ethnic/cultural, or socioeconomic factors [3], but males are affected two-to-five times more often than females [1,4].

Autism is usually diagnosed in early life, based on the delayed development of interpersonal relationships between affected individuals and their parents, siblings, and other children. By the age of three years, deficits in two major domains, i.e., social communication/social interaction and restricted/repetitive interests usually develop [5,6]. The clinical picture may vary from patient to patient by the presence of various symptoms, their frequency and severity. Typical features of autism are inhibited social initiation and response, impaired verbal communication, problems with social awareness and insight, as well as with the broader concept of social relationships [5]. Autistic children make poor eye contact with people, lack play skills, dislike being held or even touched, and overall have problems making friends. Verbal communication is limited with the possible loss of previously known words. Problems expressing needs and desires or answering questions are present. Other features of autism are preoccupation with objects or topics, and stereotyped or repetitive speech, motor movements (e.g., body rocking, hand flapping), and play with (or use of) objects. Rituals are common in autism and individuals are resistant to changes in routines. Hyper- or hypo-reactivity to sensory input (touch, sound, or light) or unusual interest in sensory aspects of the environment may be observed as well [5].

Research over the past few decades has uncovered an extreme complexity of the etiopathogenesis of autism, suggesting a complex genetic basis of the disorder [7,8,9], involvement of multiple brain regions [6,10] and different neurotransmitter systems [4,11], and a variety of prenatal and perinatal risk factors [12]. Despite major advances in genetics [8], neurobiology, and recognition of the environmental factors, the molecular mechanisms underlining the development of autism remain unclear. Consequently, there is no cure for autism [11,13], even if several behavioral and educational programs have been suggested for treatment [4,14]. The only available pharmacological approaches are aimed at alleviating the associated symptoms to facilitate easier societal integration [13,15].

Recent research suggests a possible role of oxidative stress in the etiopathology of autism, resulting from an imbalance between the production and elimination of reactive oxygen species [16,17,18,19,20,21,22,23,24,25,26,27,28]. Levels of glutathione (GSH), a major antioxidant in the brain, have been shown to be reduced in patients with autism [13,18,22,25,29,30,31,32,33,34,35,36,37,38,39]. Altered glutathione redox status (GSH/oxidized glutathione (GSSG)) has therefore been proposed as a possible therapeutic approach [13]. Moreover, several agents increasing the level of GSH have been reported to improve the status of autistic patients [29,40,41].

Methylcobalamin has been suggested to have an impact on the redox status and consequently on the clinical signs of autism. Indeed, a few studies have shown that 8–12 weeks of methylcobalamin administered intramuscularly three times a week either alone or with an oral low-dose of folic acid given twice a day in children with autism led to increased plasma concentrations of GSH and GSH/GSSG [31,42,43,44]. Moreover, clinical improvement was observed in some patients.

We followed up on these studies and examined the effect of methylcobalamin on the behavioral and psychological status of patients with autism during a prolonged, six-month treatment period. To facilitate the practicality and compliance of supplement administration, we developed a novel syrup form of methylcobalamin. Our results suggest that six months of methylcobalamin treatment has a positive impact on the cognitive, social, behavioral, and communication skills of patients with autism, which is associated with changes in the glutathione redox status.

## 2. Materials and Methods

### 2.1. Participants

Patients (male, *n* = 20; female, *n* = 5) of 4–20 years old with a primary diagnosis F84.0 were enrolled in the study. Diagnosis was determined by clinical psychiatrists following the 10th ICD Classification of Mental and Behavioral Disorders, WHO, 2003 and verified by two diagnostic methods, the Autism Diagnostic Interview-Revised (ADI-R) [45] and Childhood Autism Rating Scale (CARS-2) [46]. Exclusion criteria included a previous diagnosis of Asperger syndrome, high-functioning autism, epilepsy, pharmacotherapy affecting the CNS, co-morbidities, and infection. All participants attended specialized autistic centers five days a week.

The biochemical parameters before the methylcobalamin treatment were compared with healthy relatives (*n* = 23) and patients with Asperger syndrome (*n* = 7) of comparable ages.

### 2.2. Intervention 

A syrup form of methylcobalamin was developed for this study. Methylcobalamin (500 µg/5 mL) was administered orally within a 200-day period daily in the morning after the first meal. The dose was estimated based on the recommendation of the European Food Safety Authority. Prior to conducting the study, the project was discussed with the State Institute for Drug Control, which is the national regulatory body responsible for clinical trial approval, who concluded that our project involving a dietary supplement (a form of vitamin B 12) did not require clinical trial registration but instead should be approved by an ethical committee. Therefore, all procedures and the signed informed consent form were submitted to and approved by the Ethical Committee for Biomedical Research of the Faculty of Pharmacy, Comenius University in Bratislava and were performed in accordance with the ethical standards laid down in the 1964 Declaration of Helsinki and its later amendments. The parents of all minors gave their informed consent prior to their inclusion in the study. Details that might disclose the identity of the subjects under study were omitted.

### 2.3. Clinical and Psychological Evaluation of the Treatment Effect

The clinical and psychological profile was evaluated before methylcobalamin administration (d0), at day 100 (d100), and at day 200 (d200) of the treatment. 

Two diagnostic scales, ADI-R [45,47] and CARS-2 [46], commonly used in clinical practice were employed. Moreover, to further assess autistic characteristics, an assessment scale with 54 items was developed specifically for the purpose of this study (Appendix A) in order to address the main social (13 items), communication (10 items), behavioral (14 items), and cognitive (17 items) features that are known to be affected in patients with autism. The design was based on the criteria for childhood autism reported in the accepted classification systems (ICD-10, 2003; DSM-4, 1994) and accepted diagnostic methods such as ADI-R [45,47], CARS-2 [46], Autistic Continuum [48], and the clinical praxis. This supplementary assessment scale had no influence on the clinical psychiatrists’ diagnoses.

The social, communication, behavioral, and cognitive profile was evaluated by three trained psychologists and by the parents. The absence of an examined assessment scale feature was marked as “0” and the presence of a feature was marked as “1”. In order to introduce the polarity of the observed change, values were recoded. Positive features that ameliorated clinical and psychological status (in Appendix A marked black) remained unaffected by recoding. The presence of an adverse (negative) feature (in Appendix A gray) was recoded 1→0 and 0→1.

For each patient, the sum of all values (total ranking) and the sum of the values characterizing the particular feature group (cognitive, social, behavioral, and communication partial rankings) were calculated at d0, d100, and d200. To evaluate the change in the presence or absence of the feature at d0, d100, and d200, an absolute progress (d200-d0) and a relative progress ((d200-d0)/d0) were calculated for total and partial rankings. In order to evaluate the appearance or a disappearance of a feature during the treatment period, the values at d0, d100, and d200 were compared and each feature was recoded by means of polarity: positive change, 0→1; unaffected feature, 0→0 or 1→1; and negative change, 1→0.

### 2.4. Biochemical Analysis

The levels of GSH, GSSG, vitamin B12, homocysteine, and cysteine were determined in the plasma at d0, d100, and d200. Plasma was extracted from fresh venous blood collected into EDTA-treated tubes. Plasma used for GSSG and GSH analyses was deproteinated with 10% metaphosphoric acid at 4 °C for 10 min, and centrifuged at 4010 rpm for 10 min. The supernatant was flash-frozen and stored at −80 °C until later analysis. Plasma used for homocysteine and cysteine analyses was flash-frozen and stored at −80 °C until later analysis. Plasma level of vitamin B12 was determined in a commercial diagnostic laboratory by their routinely used procedures. GSH and GSSG levels were determined in deproteinated plasma by HPLC following a previously published protocol [49]. Briefly, glutathione reacted with orthophthaldehyde to form a stable, highly fluorescent tricyclic derivate (GSH at pH 8, GSSG at pH 12). During the measurement of GSSG, GSH was complexed to N-ethylmaleimide. Analytical separation was performed at 37 °C on a liquid chromatograph (Agilent Technologies HP 1050 Series; Waldbronn, Germany) with a fluorescence detector. The system consisted of an analytical column (Zorbax Eclipse XDB-C18, 150 mm × 4.6 mm) with a sorbent particle size of 5 μm and a precolumn (Zorbax Eclipse XDB-C18, 12.5 mm × 4.6 mm) with the same particle size (Agilent Technologies, Halbron, Germany). The mobile phase was a mixture of methanol and 25 mM sodium hydrogenphosphate (15:85, *v/v*, pH 6.0) with a flow rate of 0.5 mL/min and an injection volume of 100 µL. Fluorescent detection wavelengths were set at 350 nm for excitation and 420 nm for emission. GSH and GSSG concentrations were determined from the calibration curves with linearity in the whole tested range (GSH: 0.1–4.7 µmol/L, R^2^ = 0.999 and 4.7–66.6 µmol/L, R^2^ = 0.999; GSSG: 0.6–12.5 µmol/L, R^2^ = 0.998 and 12.5–100 µmol/L, R^2^ = 0.999). The limit of quantification was 0.1 µmol/L for GSH and 0.6 µmol/L for GSSG. The recoveries were 90.3% for 0.5 µmol/L GSH, 92.5% for 1.9 µmol/L GSH, 90.8% for 0.8 µmol/L GSSG and 91.0% for 1.9 µmol/L GSSG. 

The protocol of Vilaseca and colleagues [50] was used to determine the plasma levels of homocysteine and cysteine. Briefly, samples were reduced with tri-n-butylphosphine and, after protein precipitation, derivatization with ammonium 7-fluorobenzo-2-oxa-1,3-diazole-4-sulphonate was performed. The HPLC separation and fluorescence detection was performed at 25 °C with the same parameters as described above for the GSH and GSSG detection. Isocratic separation of homocysteine and cysteine was carried out with the mobile phase acidified with sodium acetate trihydrate (71.65 mM CH_3_COONa·3H_2_O, pH adjusted to 4 with glacial acetic acid); after pH adjustment, 20 mL of methanol was added to 1 L of the solution. The flow rate of the mobile phase was 0.6 mL/min, and the injection volume was 30 µL. The excitation and emission wavelengths were set at 385 nm and 515 nm, respectively. Cysteine and homocysteine concentrations were determined from the calibration curves. Linearity was observed in the whole tested range (cysteine: 20–200 µmol/L, R^2^ = 0.999; homocysteine: 2.2–200 µmol/L, R^2^ = 0.999). The limit of quantification was 20 µmol/L and 2.2 µmol/L, respectively. The recoveries were 95.9% for 30 µmol/L and 97.8% for 100 µmol/L of cysteine and 96.4% for 2.5 µmol/L and 98.1% for 5.0 µmol/L of homocysteine. 

### 2.5. Statistics

The SAS Education Analytical Suite (version 9.3, 2016) was used to analyze the data. The non-parametric Friedman test was used for the analysis of the dependent samples of participants in general (i.e., biochemical data or total ranking) and within and between specific clinical and psychological feature groups (i.e., partial ranking). The non-parametric post hoc Wilcoxon signed-rank test was applied for two dependent samples, with p0–100 describing the difference in the values obtained at d0 and d100, p0–200 between d0 and d200, and p100–200 between d100 and d200. The evaluations (total ranking in d0 and total change in d200) obtained from the parents were compared with the evaluations obtained from the psychologists. The supposed relationship between these two assessments was processed by the Passing–Bablok method, indicating a full agreement in the analyzed data (r = 0.9775, with practically zero intercept and unity slope, *p* < 0.0001) (Appendix A). Analyses of the variance components using the Bayesian estimate confirmed (Appendix A) that the majority of the variability (97.75%) was based on the variations between individuals and not the difference in assessment by psychologists and parents (<1%). Cronbach alphas, calculated separately for each social and psychological partial ranking and the total ranking were not less than 0.81, which indicates a very good internal consistency of the used assessment scale. To investigate the association between biochemical parameters and clinical and psychological profiles, the biochemical data and clinical and psychological data were stratified by their respective medians to High (>median) and Low (≤median) subgroups. These transformed categorical data with a 2 × 2 design were analyzed using the Fisher exact test. The Fisher exact test was described as odds ratios (OR) and 95% confidence intervals (CI). Treatment time was used as a third, blocking factor for additional stratification in the Cochran–Mantel–Haenszel tests (CMH). The associations of clinical and psychological features with biochemical features were confirmed by the standard linear mixed model (with respect to patients) of the individual partial rankings or total ranking and biochemical parameters and treatment time, including the crossed effects between the time and biochemical data. Effect sizes were modeled and evaluated by the Cohen delta, indicating a no effect (<0.20) small effect (0.20–0.40), middle effect (0.41–0.80), or large effect (>0.80). In all data analyses, calculated *p* values less than 0.05 were considered as statistically significant.

## 3. Results

### 3.1. A Syrup Form of Methylcobalamin

In order to facilitate the administration, methylcobalamin was prepared in syrup form for the needs of this project. Indeed, this galenic formulation led to good patient compliance, and no signs of stressed behavior were observed just before, during, or after methylcobalamin administration. The patients showed no resistance to the dosing and willingly took the whole tested dose of methylcobalamin. 

### 3.2. Clinical and Psychological Features 

Methylcobalamin administration did not result in any radical changes in the patients’ status. Nevertheless, an improvement was reported by the parents and examining psychiatrist following the detailed examination scale that was developed and validated for the use of this project. We observed statistically significant differences (*p* < 0.0001) in the total ranking values (Figure 1) of the clinical and psychological features between any pair of the values obtained at d0, d100, and d200 (p0–100 = 0.0002, p100–200 = 0.0022, p0–200 < 0.0001), while the total ranking value increased by 50% at the end of the treatment period (Table 1). 

Moreover, we determined statistically significant differences between the partial ranking values in all feature groups (Table 1). The highest increase in the absolute progress values as well as the gain of treatment normalized for scale length was observed in the social features (*p* < 0.0001; p0–100 < 0.0001, p100–200 = 0.0369, p0–200 < 0.0001), followed by the cognitive functions (*p* < 0.0001; p0–100 = 0.00082, p100–200 = 0.0140, p0–200 < 0.0001), the behavioral features (*p* = 0.0045; p0–100 = 0.3221, p100–200 = 0.0014, p0–200 = 0.0010), and the communication skills (*p* < 0.0001; p0–100 = 0.0002, p100–200 = 0.1091, p0–200 < 0.0001). The absolute and relative progress treatment effects assessed by the Cohen delta showed the highest effect size in the partial ranking value for social features, followed by cognitive, communication, and behavioral skills.

Methylcobalamin treatment led to changes in the observed clinical and psychological features and the number of responding subjects (represented by the slope, Figure 2), while these changes were more related to the amelioration of the clinical and psychological status than its regression (documented by the areas under the curve, AUC). We observed that in 48% of participants, at least 10 features were changed at d100 of the methylcobalamin treatment, while in the majority of them there was an improvement (92%) and only one individual responded by a deterioration. In 20% of participants, at least 20 features were changed and in one patient as many as 27 features were altered, and all of the changed features showed improvements. Prolonged treatment (d200) enhanced the response in regard to the number of responsive subjects as well as the number of clinical and psychological features that showed improvement. A total of 68% of subjects showed changes in at least 10 features. Improvement of the features occurred in 64% and worsening was observed in one patient. In 32% of the enrolled subjects, at least 20 features changed; all of them were represented by an amelioration. One patient showed improvement in as many as 32 features (out of 54 features in total). 

As mentioned above, all four clinical and psychological categories were affected during the duration of the treatment (Table 1), with the greatest impact on social features (change of the ranking variable, RV = 4.12 ± 3.60) and the least impact on communication features (change of the ranking variable, RV = 1.40 ± 1.63). In the social feature category (Figure 3a), motivation to verbalization (SCL.11), feedback on information from another person (SCL.12), and respecting communication partner (SCL.13) were improved in more than half of participants. At least 40% of patients showed improvements in social features such as interest in physical contact (SCL.2), active interest in peers (SCL.4), social destructivity (SCL.6), and cooperative play (SCL.10). In other categories, a change in at least 40% of the participants was observed in destructivity (BHV.7) and psychomotor sensitivity (BHV.12) (behavioral features, Figure 3b), appropriate use of verbalization (CMN.10) (communication features, Figure 3c), simple symbolic behaviors (CGN.1), reaction to stimuli from another party (CGN.5), sensitivity to sounds (CGN.8), academic abilities (CGN.15), the use of knowledge (CGN.16), and imitation (CGN.17) (cognitive features, Figure 3d). The most deteriorating changes were observed in the group of behavioral features (Figure 3b) where all features contained a small percentage of participants (≤16%) that regressed at some point. The most frequent features that showed regression (present in more than 20% of participants) were reaction to emotional behavior of another person (d100 and d200; communication features, Figure 3c), fascination by stimuli/objects, and problems with shifting attention from stimuli, activity (d200; cognitive features, Figure 3d). 

### 3.3. Biochemical Parameters

We assessed multiple biochemical parameters in the plasma of the patients collected on d0, d100, and d200, including the levels of GSH, GSSG, vitamin B12, homocysteine, and cysteine (Table 2).

The level of GSH in autistic patients (*n* = 25) was more than 3-fold lower than the one observed in the control group (0.36 ± 0.40 µmol/L vs. 1.24 ± 2.20 µmol/L, *n* = 23; *p* = 0.0539, Appendix A), whereas the levels of GSSG (1.48 ± 0.54 µmol/L vs. value 1.34 ± 0.97 µmol/L, *n* = 23; *p* = 0.5523), GSH/GSSG (0.26 ± 0.26 vs. 1.24 ± 2.87, *n* = 23; *p* = 0.0967), cysteine (95.29 ± 39.79 µmol/L vs. value 95.44 ± 40.69 µmol/L, *n* = 22; *p* = 0.9903), and homocysteine (3.69 ± 1.21 µmol/L vs. value 3.55 ± 1.09 µmol/L, *n* = 22; *p* = 0.6738) were comparable. For comparisons, in the group of patients with Asperger syndrome (*n* = 7), the level of GSH was 0.43 ± 0.30 µmol/L, GSSG 1.90 ± 1.71 µmol/L; GSH/GSSG 0.34 ± 0.27, cysteine 76.66 ± 50.64 µmol/L, homocysteine 30.18 ± 1.54 µmol/L. 

The methylcobalamin treatment had an important impact on the GSH levels in autistic patients and increased more than 2-fold at d200, with a significant increase observed from d100 (*p* < 0.0001; p0–100 = 0.0774, p100–200 = 0.0018, p0–200 < 0.0001). Methylcobalamin did not change the oxidized form of GSSG within 200 days, despite a slight increase at d100 (*p* = 0.2869; p0–00 = 0.0209, p100–200 = 0.7859, p0–200 = 0.0859). However, methylcobalamin treatment significantly increased the overall oxidative status, expressed as GSH/GSSG, reaching a 2,5-fold increase from d0 to d200 (*p* = 0.0004; p0–100 = 0.0971, p100–200 = 0.0060, p0–200 < 0.0001).

Treatment with methylcobalamin led to a 1,5-fold increase in the plasmatic levels of vitamin B12 (*p* < 0.0001; p0–100 = 0.0023, p100–200 = 0.2223, p0–200 < 0.0001). In the case of homocysteine, we observed no statistically significant changes (*p* = 0.1926; p0–100 = 0.6127, p100–200 = 0.0720, p0–200 = 0.2453). Methylcobalamin treatment did not affect cysteine levels at d100 of the treatment, but at d200 we observed a significant increase (*p* = 0.0002; p0–100 = 0.4805, p100–200 = 0.0023, p0–200 < 0.0067).

### 3.4. Clinical and Psychological Features vs. Biochemical Parameters

The clinical and psychological profiles of the patients were not associated with the levels of cysteine and homocysteine (Table 3). While there was an association between vitamin B12 and the overall clinical and psychological profiles, no significant association was observed when analyzed for a particular category.

The strongest association (expressed by odds ratios) was observed with the level of GSH, where the total and partial ranking of the clinical and psychological features showed significance with the Fisher exact test. However, the association of GSH with the behavioral features was not affected by time, and there was only a marginal association between GSH and the total ranking when stratifying the data by time (as assessed by the CHM test). Indeed, this strong association was confirmed in the linear mixed model of original data also (Table 4).

These results reflect the associations with the oxidative status, where all analyzed rankings except for the communication partial ranking showed significant associations with the levels of GSH/GSSG (by the Fisher exact test), although the association with the cognitive partial ranking did not depend on time. No association between the clinical and psychological status and GSSG was observed.

The linear mixed model of original data showed comparable results (Table 4).

## 4. Discussion

### 4.1. Syrup Form of Methylcobalamin Enhanced Patient Compliance 

Methylcobalamin has been tested in patients with autism spectra disorder and positive effects have been reported [31,42,43,51]. In those studies, however, methylcobalamin was administered as an injection, twice a week, which is rather stressful for the patients and their parents and requires logistical arrangements for doctor visits. Here we introduced a novel galenic form of methylcobalamin to be offered to the autistic patients. We report excellent patient compliance with no incidence of dose refusal, but rather the patients were very keen on receiving the dose, which eliminated stress for the patients and for the parents. Additionally, a syrup form enables administration of the dose at home and thus eliminates the need for frequent doctor visits. 

### 4.2. Orally Administered Methylcobalamin Alleviated Autistic Symptoms

Previous work has shown that the subcutaneous administration of methylcobalamin enhanced the clinician-rated symptoms of autism [31,42,43,51]. There have been no published studies with orally administered methylcobalamin, probably due to the complicated administration in autistic patients as well as limited bioavailability. A prior study reported 9.7 µg of cobalamin to be absorbed from a single 500 µg oral dose, which accounts for only 1.9 % of the dose [52]. Absorption of cobalamin (vitamin B12) in the gastrointestinal tract is almost entirely dependent on the intrinsic factor. It decreases drastically once the capacity of the intrinsic factor is met, i.e., at a dose of approximately 1–2 µg. Studies on patients with pernicious anemia or gastroduodenal resection have suggested that a small amount of absorption is provided via passive diffusion, accounting for about 1.2% of the total absorption [52,53]. High enough doses of orally administered vitamin B12 could therefore provide sufficient absorption. Indeed, it has been shown that hematological and neurological responses to high oral doses of vitamin B12 (i.e., 1000–2000 µg) were comparable to parenteral administration [54,55]. 

Here we report for the first time that an oral application of high doses of methylcobalamin had beneficial effects in patients with autistic spectra disorder, which is consistent with previously described subcutaneous injection delivery [31,42,43]. Supplementation with methylcobalamin at a dose of 500 µg per day increased vitamin B12 plasma levels 1.5 times. The overall clinical and psychological total ranking of the patients was associated with vitamin B12 levels and improved with the duration of the treatment period. Likewise, the number of improved features increased throughout the treatment. Within the clinical, social, and behavioral categories, the most affected features were those enabling the interaction with others, including motivation and appropriate use of verbalization, understanding simple information and feedback on information from another person, respecting communication partner, cooperative play, and interest in physical contact. During the progress of the study, we observed that methylcobalamin treatment eliminated nocturnal enuresis in three of the patients enrolled in the study, while at least in one of them it was achieved due to the treatment [56]. Administration of methylcobalamin in an 18-year-old male resulted in a gradual disappearance of nocturnal enuresis during the 6-week interval. Interruption of the therapy led to the reappearance of bed wetting, but with a lower frequency. Based on changes in the psychological and behavioral profile of the patient, especially increased interest in social interaction and the family environment, it was suggested that the reason for this phenomenon was an increased awareness of the full bladder. Indeed, in this larger group studied here, we observed a significant enhancement of interest and reaction to stimuli from the environment. 

### 4.3. Mechanism of Action of Methylcobalamin

Methylcobalamin is the predominant physiologically active form of cobalamin (vitamin B12) that has a crucial role in physiological cell metabolism. As a co-factor of the cytoplasmic methionine synthase, it participates in the synthesis of methionine by transferring the methyl group from methyltetrahydrofolate to homocysteine. It thus protects the cells from vasculotoxic and neurotoxic levels of homocysteine, indirectly controls methionine-dependent methylation capacity and glutathione-controlled redox homeostasis, and via the folate cycle, DNA replication and repair [43,53,57,58]. 

The ability of methylcobalamin to influence methionine metabolism and improve cellular methylation capacity, mitochondrial dysfunction, and oxidative stress has been linked with its therapeutic potential in patients with autism [13,42,43,44]. An observation that supplementation with vitamins B6 and B12 (with or without folic acid) decreases levels of urinary homocysteine in autistic patients supports intervention at the level of methionine metabolism [59]. 

Hyperhomocysteinemia negatively correlates with glutathione peroxidase activity in the erythrocytes of autistic patients, indicating perturbations in the methionine cycle and redox homeostasis [60]. Redox homeostasis is a balance between the production of oxidants and endogenous antioxidant defense mechanisms and its disturbance leads to oxidative stress that has deteriorating effects on cells [16]. Oxidative stress, clinically defined by a decrease in the GSH/GSSG ratio, has been described in many neurobehavioral disorders, including autism [13,37,39,61]. Indeed, as mentioned above, multiple studies, including ours, reported lower GSH levels and a lower GSH/GSSG ratio in patients with autism in comparison to healthy controls [16,18,23,26], which were elevated during methylcobalamin treatment [31,37,44]. Intramuscular injection of methylcobalamin alone or with orally administered folic acid for a 3-month period caused an increase in the GSH levels and the GSH/GSSG ratio [31,44,62]. Interestingly, contrary to the combination of methylcobalamin with low-dose folic acid, a high-dose folic acid treatment did not show any major significant effect on redox regulation or methylation pathways. A lack of pre-screening for low baseline GSH/GSSG has been speculated as a contributing factor for this discrepancy [62], but we argue that the effect may be independent of the folic acid and may rather be set by the methylcobalamin. In our 200-day study, methylcobalamin syrup increased cysteine plasma levels while homocysteine levels were unaltered. Moreover, we observed a shift towards the reduced form of glutathione that approached the levels of healthy controls. Importantly, the clinical and psychological profiles of the patients showed a strong association with the GSH levels and GSH/GSSG ratio. 

No such changes were, however, observed during a randomized placebo-controlled 8-week study of methylcobalamin injection in autistic patients. This study reported no change in redox balance, but instead a correlation between clinical improvement (Clinical Global Impressions–Improvement score) and increases in methionine levels, decreases in S-adenosylhomocysteine levels, and improvements in the S-adenosylmethionine/S-adenosylhomocysteine ratio were reported [43]. Epigenetic alterations (DNA damage and hypomethylation) have been observed in autistic children but not paired siblings or controls [26]. Autistic patients have been shown to have 50% of the index of cellular methylation capacity of the healthy age-matched controls [23], which gives a potential for methylation improvement. Although enhancement of methionine metabolism and cellular methylation capacity after methylcobalamin supplementation have not been described, this remains an interesting avenue for future investigation. 

### 4.4. Oral Methylcobalamin Is Safe 

Vitamin B12 is considered to be a non-toxic compound and no tolerable upper intake level has been set by the European food safety authority [63] and the US Food and Nutrition Board [64]. In the case of oral administration of high doses of methycobalamin, no severe adverse effects were noted. We observed slightly worse psychological and behavioral features in a few patients four to six weeks from the start of the treatment, but they did not require discontinuation of the treatment. Similar observations have been reported after subcutaneous injection of methylcobalamin (75 µg/kg) twice a week and oral folinic acid 400 µg/kg twice a day [42], and no statistically significant increase in adverse events was seen among methylcobalamin-treated (75 µg/kg, s.c., every three days) children enrolled in the randomized controlled trial [43].

## 5. Conclusions

In conclusion, the syrup form of methylcobalamin enables easy administration in the comfort of the home and shows good patient compliance. A 200-day methylcobalamin supplementation improved the clinical state of patients with autism, with the biggest impact on social interaction. The clinical changes were strongly associated with redox status measures. A relatively low incidence of adverse effects was observed during the study. The results of our study suggest syrup methylcobalamin supplementation to be a potential intervention in patients with autism. 

## 6. Patents

There is a Utility Model (#7302) resulting from the work reported in this manuscript.

## Figures and Tables

**Figure 1 nutrients-14-02035-f001:**
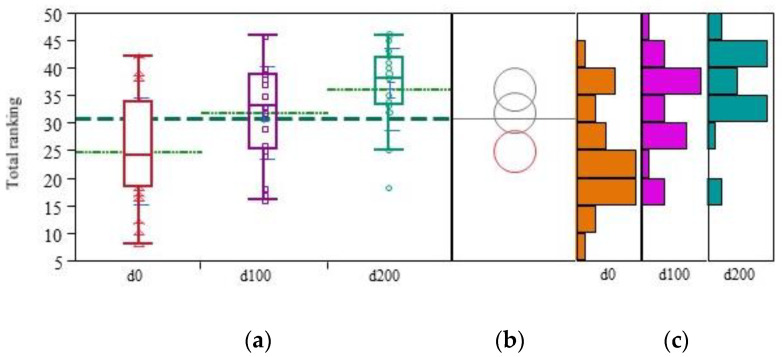
Overall enhancement of the clinical and psychological status. Boxplot diagrams at different treatment periods (**a**), Dunnett’s test against the control at d0 (red circle) (**b**), and distribution histograms of changes during the treatment (**c**) of the overall enhancement of the clinical and psychological status of the patients with autism during the methylcobalamin supplementation (d0—before the treatment, d100—at day 100, d200—at day 200 of the treatment).

**Figure 2 nutrients-14-02035-f002:**
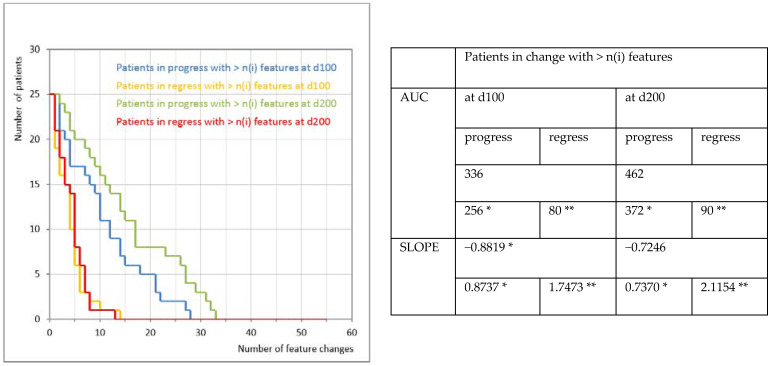
Clinical and psychological feature dynamics of patients. Positive (green) and negative (orange, red) feature changes at day 100 (d100, dotted lines) and day 200 (d200, solid lines) of the methylcobalamin treatment. * More positive or bigger value is better. ** Less positive or lesser value is better.

**Figure 3 nutrients-14-02035-f003:**
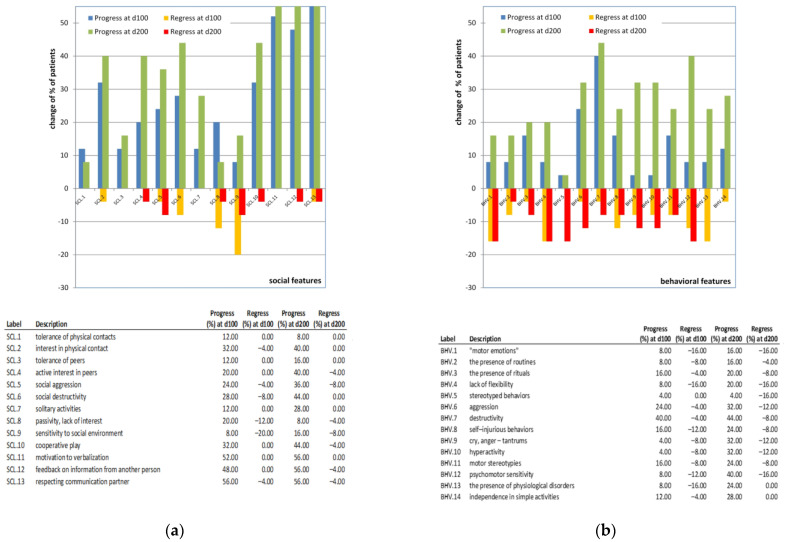
Progression and regression of the clinical and psychological features. Changes observed in the social (**a**), behavioral (**b**), communication (**c**), and cognitive (**d**) domains at d100 and d200 of the methylcobalamin treatment. Besides the routinely examined adverse effect of the treatment (Appendix A), additional adverse effects were reported in some patients by the psychologists and parents. Among them, sleep distortion (*n* = 5), behavioral disturbance towards enhanced affectivity (e.g., self-destructive behavior) (*n* = 4), increased hyperactivity (*n* = 6), and licking or putting in mouth inappropriate objects (*n* = 3) were observed. These adverse behaviors usually occurred around four to six weeks after the treatment started.

**Table 1 nutrients-14-02035-t001:** Total and partial ranking values (Rv) of the clinical and psychological assessments during the study (d0—before the treatment, d100—at day 100 of treatment, d200—at day 200 of treatment) and evaluation of the overall absolute and relative change (d200 vs. d0 values) explained by Cohen delta, and normalized gain of the treatment.

Ranking Variable	Rv(d0) (±s_R_)	Rv(d100) (±s_R_)	Rv(d200) (±s_R_)	Rv(d200-d0) (±s_R_)	Absolute Treatment Effect Cohen Delta	Rv(d200-d0)/R(d0) (±s _R_)	Relative Treatment Effect Cohen Delta	Gain of Treatment Normalized for Scale Length (%)
Social (scale length = 13)	5.16 (±2.94)	8.2 (±2.65)	9.28 (±2.69)	4.12 (±3.60)	1.08 ^+++^	1.01 (±1.06)	0.89 ^+++^	31.69
Behavioral (scale length = 14)	6.16 (±3.30)	6.72 (±2.89)	8.36 (±2.50)	2.20 (±4.00)	0.51 ^++^	1.01 (±1.99)	0.47 ^++^	15.71
Communication (scale length = 10)	4.44 (±1.66)	5.36 (±2.02)	5.84 (±1.68)	1.40 (±1.63)	0.80 ^+++^	0.44 (±0.60)	0.68 ^++^	14.00
Cognitive (scale length = 17)	9.08 (±3.09)	11.6 (±2.84)	12.64 (±2.14)	3.56 (±3.14)	1.05 ^+++^	0.57 (±0.67)	0.78 ^++^	20.94
Total (scale length = 54)	24.84 (±9.69)	31.88 (±8.29)	36.12 (±7.43)	11.28 (±10.84)	0.97 ^+++^	0.70 (±0.89)	0.73 ^++^	20.89

Note: Rv(d200-d0) and Rv(d200-d0)/d0 are means of differences obtained for individual patients. Patients missing either d0 or d200 values were not included in the calculation. ++ Medium treatment effect. +++ Large treatment effect.

**Table 2 nutrients-14-02035-t002:** Biochemistry parameters of the treatment group (Bio) during the study (d0—before the treatment, d100—at day 100 of treatment, d200- at day 200 of treatment), and the evaluation of the overall absolute and relative changes (d200 vs. d0 values) explained by Cohen delta.

Parameter	Bio(d0) (±s_R_)	Bio(d100) (±s_R_)	Bio(d200) (±s_R_)	Bio(d200-d0) (±s_R_)	Absolute Treatment Effect Cohen delta	Bio(d200-d0)/Bio(d0) (±s _Bio_)	Relative Treatment Effect Cohen Delta
GSH	0.36 (±0.40)	0.48 (±0.44)	0.78 (±0.65)	0.42 (±0.66)	0.58 ^++^	2.74 (±3.48)	0.73 ^++^
GSSG	1.48 (±0.54)	1.57 (±1.34)	1.36 (±0.66)	−0.11 (±0.73)	−0.14	0.05 (±0.69)	0.06
GSH/GSSG	0.26 (±0.26)	0.39 (±0.33)	0.66 (±0.59)	0.40 (±0.51)	0.73 ^++^	3.15 (±4.26)	0.69 ^++^
Vit. B12	408.48 (±474.9)	601.33 (±404.92)	628.64 (±209.95)	211.33 (±423.46)	0.46 ^++^	1.03 (±0.93)	1.02 ^+++^
Homocysteine	3.69 (±1.21)	3.47 (±0.87)	4.02 (±1.17)	0.29 (±1.14)	0.23 ^+^	0.14 (±0.38)	0.34 ^+^
Cysteine	95.29 (±39.79)	85.35 (±32.01)	115.68 (±30.58)	19.69 (±44.40)	0.42 ^++^	0.43 (±0.91)	0.44 ^++^

Note: Rv(d200-d0) and Rv(d200-d0)/d0 are means of differences obtained for individual patients. Patients missing either d0 or d200 values were not included in the calculation. + Small treatment effect. ++ Medium treatment effect. +++ Large treatment effect.

**Table 3 nutrients-14-02035-t003:** Association between the biochemical parameters and the psychological or clinical data. Original rankings data were stratified by the respective medians to Hi (>median)) and Low (<= median) subgroups. For these transformed categorical data with a 2 × 2 design the Fisher exact test was applied, and the respective effects are described as odds ratios and 95% confidence intervals (CI). Time-related data was used as a third, blocking factor in the Cochran–Mantel–Haenszel tests (CMH). Statistical significance for *p* < 0.05 is indicated with the * asterisk symbol.

Biochemical Parameter	Statistics Descriptor	Social Ranking Statistics	Behavioral Ranking Statistics	Communication Ranking Statistics	Cognitive Ranking Statistics	Total Ranking Statistics
**GSH**	Fisher exact OR (CI) CMH	0.0002 * 7.4388 (2.5967; 21.3103) 0.0040 *	0.0381 * 2.8000 (1.0945; 7.1629) 0.2225	0.0091 * 3.8961 (1.4748; 10.2927) 0.0338 *	0.0002 * 7.0000 (2.4993;19.6048) 0.0148 *	0.0056 * 3.9487 (1.5073; 10.3441) 0.0515
**GSSG**	Fisher exact OR (CI) CMH	0.6440 0.7519 (0.3018; 1.8732) 0.8187	0.8188 0.8571 (0.3459; 2.1236) 0.9052	0.2412 1.9000 (0.7458; 4.8402) 0.0244 *	1.0000 0.9418 (0.3803; 2.3324) 0.3369	1.0000 0.9580 (0.3864; 2.3752) 0.6335
**GHS/GSSG**	Fisher exact OR (CI) CMH	0.0023 * 4.8148 (1.7853; 12.9855) 0.0297 *	0.0055 * 3.9808 (1.5216; 10.4151) 0.0307 *	0.1074 2.1991 (0.8620; 5.6103) 0.3694	0.0025 * 4.6429 (1.7429; 12.3678) 0.0693	0.0026 * 4.4722 (1.6922; 11.8191) 0.0199 *
**B12**	Fisher exact OR (CI) CMH	0.0519 2.7692 (1.0243; 7.4865) 0.5640	0.6280 1.3333 (0.5087; 3.4945) 0.5955	0.1411 2.2727 (0.8473; 60.959) 0.6156	0.0519 2.7692 (1.0243; 7.4865) 0.9010	0.0280 * 3.2168 (1.1811; 8.7608) 0.2469
**HCY**	Fisher exact OR (CI) CMH	0.6264 0.7143 (0.2738; 1.8632) 0.4040	0.4664 1.6071 (0.6158; 4.1938) 0.4375	0.6245 1.3866 (0.5282; 3.6391) 0.4836	1.0000 0.8947 (0.3449; 2.3206) 0.6624	0.6273 1.4166 (0.5445; 3.6857) 0.5156
**CYS**	Fisher exact OR (CI) CMH	0.6324 0.7500 (0.29042; 1.9368) 0.0817	0.6329 1.3359 (0.5185; 3.4421) 0.6874	0.6279 0.735119 (0.2816; 1.9190) 0.2110	0.8112 1.1912 (0.4631; 3.0645) 0.2947	0.6324 0.7456 (0.2891; 1.9230) 0.1119

**Table 4 nutrients-14-02035-t004:** Linear mixed model effects between partial rankings (or total ranking) and monitored biochemical parameters (Bio = GSH, GSH/GSSG), treatment time (Time), and cross effects (Bio × Time). Index of determination (R^2^) and statistical significance of the whole model p(Model) and individual regression coefficients p(Intercept), p(Bio), p(Time), p(Bio × Time) were determined for each combination of biochemical regressor and psychological outcome parameter. Statistical significance for *p* < 0.05 is indicated with the * asterisk symbol. The related diagrams display a level of agreement between the predicted and experimental values.

Biochemical Parameter	Statistics Descriptor	Social Ranking Statistics	Behavioral Ranking Statistics	Communication Ranking Statistics	Cognitive Ranking Statistics	Total Ranking Statistics
**GSH**	R^2^ p(Model) p(Intercept) p(GSH) p(Time) p(GSH × Time)	0.7444 <0.0001 * <0.0001 * <0.0001 * 0.0144 * 0.0088 *	0.6656 <0.0001 * <0.0001 * 0.0234 * 0.0647 0.0136 *	0.8091 <0.0001 * <0.0001 * 0.0007 * 0.1131 0.0357 *	0.7344 <0.0001 * <0.0001 * <0.0001 * 0.0706 0.0113 *	0.7721 <0.0001 * <0.0001 * <0.0001 * 0.0140 * 0.0022 *
		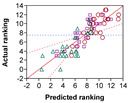	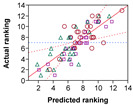	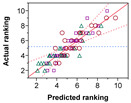	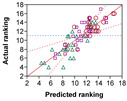	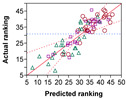
**GSH/GSSG**	R^2^ p(Model) p(Intercept) p(GSH/GSSG) p(Time) p(GSH/GSSG × Time)	0.7512 <0.0001 * <0.0001 * <0.0001 * 0.0149 * 0.0025 *	0.6660 <0.0001 * <0.0001 * 0.0499 * 0.0630 0.0115 *	0.8015 <0.0001 * <0.0001 * 0.0007 * 0.4341 0.0962 *	0.7010 <0.0001 * <0.0001 * <0.0001 * 0.0660 0.0115 *	0.7715 <0.0001 * <0.0001 * <0.0001 * 0.0202 * 0.0017 *
		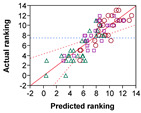	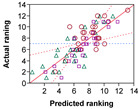	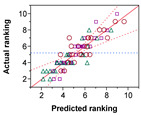	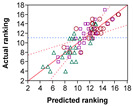	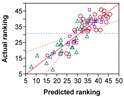

## Data Availability

Not applicable.

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
