# Peer review of "Improvement of the Clinical and Psychological Profile of Patients with Autism after Methylcobalamin Syrup Administration"

_nutrients, 2022, doi:10.3390/nu14102035_

Round 1
Reviewer 1 Report
In this article, the effect of orally administered methylcobalamin on ASD individuals has been studied. As discussed, several previous studies have shown that methylcobalamin injection improves ASD phenotypes, but there has been no study which examined the effect of oral methylcobalamin treatment. The authors developed the syrup form of methylcobalamin which showed good patient compliance and examined the effect of the treatment on clinical phenotypes, glutathione redox-related biochemical measures, and the correlation between clinical and biochemical parameters. The chronic treatment with methylcobalamin demonstrated an overall improvement in ASD phenotypes, particularly in deficits in social interaction. Ameliorations in clinical phenotypes were strongly associated with the reduction in glutathione level and in reduced/oxidized glutathione ratio suggesting that the effect is through correcting these metabolic pathways. While the placebo effect cannot be eliminated, these results indicate that oral methylcobalamin treatment may be beneficial and have the potential as a therapeutic option for ASD individuals. The manuscript is well written, the logic is straightforward, the interpretation of the data is appropriate, and the findings are novel and significant. This study likely motivates researchers in the field to conduct larger scale studies and clinical trials. I have a few minor comments.
Line 124 : In pediatric cases, maybe parental informed consent was taken?
Line 209 and Line 210 : Figure S1A and S1B may be typos for S2A and S2B.
Line 269 : In Fig2, the colors representing "Patients in regress with > n(i) features at d100" and "Patients in progress with > n(i) features at d200” may be difficult to be disambiguated. It may be better to use consistent colors used in Figure 3 for more clarity.
Line 326 – 327 : It may be better the authors also present the GSH level in box and/or scatter plot to make it clearer how the data distribution looks. While the difference is not statistically significant, it looks robust compared to previous works reporting modest alterations.
Author Response
Line 124 : In pediatric cases, maybe parental informed consent was taken?
Indeed, the informed consent was provided by the parents. This is now clarified in the methods.
Line 209 and Line 210 : Figure S1A and S1B may be typos for S2A and S2B.
Thanks for finding the discrepancy between the text and the supplement. We have changed it in the legend and the title of the supplement to Figure S1A and S1B.
Line 269 : In Fig2, the colors representing "Patients in regress with > n(i) features at d100" and "Patients in progress with > n(i) features at d200” may be difficult to be disambiguated. It may be better to use consistent colors used in Figure 3 for more clarity.
Thank you for pointing out. This was a formatting issue that we have resolved and the colors are now consistent.
Line 326 – 327 : It may be better the authors also present the GSH level in box and/or scatter plot to make it clearer how the data distribution looks. While the difference is not statistically significant, it looks robust compared to previous works reporting modest alterations.
We agree with the suggestion. We included the scatter plot of the data as Figure S2.
Reviewer 2 Report
This a well-researched and well-designed study demonstrating the efficacy of methylcobalamin administration as a potential treatment for ASD. The study used state-of-the-art pre-post assessments using CARS-2 and ADRI-R. While there is a relatively low N, the study demonstrated remarkable results as shown in table and figure 1. This is an important contribution to the literature and should be replicated in a larger group, with varied doses and administration routes of methylcobalamin.
The statistics were impressively appropriate.
An additional strength of the study was the inclusion of REDOX biochemical parameters
The authors acknowledge all the important aspects of methylcobalamin metabolism in the discussion, including bioavailability of oral versus injectable does.
Line 58: “impaired non-verbal communication”. Do you mean “impaired verbal communication”
Author Response
This a well-researched and well-designed study demonstrating the efficacy of methylcobalamin administration as a potential treatment for ASD. The study used state-of-the-art pre-post assessments using CARS-2 and ADRI-R. While there is a relatively low N, the study demonstrated remarkable results as shown in table and figure 1. This is an important contribution to the literature and should be replicated in a larger group, with varied doses and administration routes of methylcobalamin.
The statistics were impressively appropriate.
An additional strength of the study was the inclusion of REDOX biochemical parameters
The authors acknowledge all the important aspects of methylcobalamin metabolism in the discussion, including bioavailability of oral versus injectable does.
We thank the Reviewer for the positive feed-back.
Line 58: “impaired non-verbal communication”. Do you mean “impaired verbal communication”
Thanks for catching the mistake that has been corrected now.